# Analysis and Evaluation on Residual Strength of Pipelines with Internal Corrosion Defects in Seasonal Frozen Soil Region

**Xiaoli Li** [1,2], **Guitao Chen** [1], **Xiaoyan Liu** [1,2,*], **Jing Ji** [1,2] and **Lianfu Han** [3,*]

1 Department of Civil Engineering, Northeast Petroleum University, Daqing 163318, China; lixiaoli@nepu.edu.cn (X.L.); chenguitao4019@163.com (G.C.); jijing@nepu.edu.cn (J.J.)
2 Hei-Longjiang Key Laboratory of Disaster Prevention and Reduction and Protective Engineering, Daqing 163318, China
3 School of Physics and Electronic Engineering, Northeast Petroleum University, Daqing 163318, China
* Correspondence: liu_xydq@126.com (X.L.); Lianfuhan@nepu.edu.cn (L.H.); Tel.: +86-139-4694-2531 (X.L.); +86-182-496-29368 (L.H.)

**Abstract:** In order to study the residual strength of buried pipelines with internal corrosion defects in seasonally frozen soil regions, we established a thermo-mechanical coupling model of a buried pipeline under differential frost heave by using the finite element elastoplastic analysis method. The material nonlinearity and geometric nonlinearity were considered as the basis of analysis. Firstly, the location of the maximum Mises equivalent stress in the inner wall of the buried non-corroded pipeline was determined. Furthermore, the residual strength of the buried pipeline with corrosion defects and the stress state of internal corrosion area in the pipeline under different defect parameters was analyzed by the orthogonal design method. Based on the data results of the finite element simulation calculation, the prediction formula of residual strength of buried pipelines with internal corrosion defects was obtained by SPSS (Statistical Product and Service Solutions) fitting. The prediction results were analyzed in comparison with the evaluation results of B31G, DNV RP-F101 and the experimental data of hydraulic blasting. The rationality of the finite element model and the accuracy of the fitting formula were verified. The results show that the effect degree of main factors on residual strength was in order of corrosion depth, corrosion length, and corrosion width. when the corrosion length exceeds 600 mm, which affects the influence degree of residual strength will gradually decrease. the prediction error of the fitting formula is small and the distribution is uniform, it can meet the prediction requirements of failure pressure of buried pipelines with internal corrosion defects in seasonally frozen soil regions. This method may provide some useful theoretical reference for the simulation real-time monitoring and safety analysis in the pipeline operation stage.

**Keywords:** internal corrosion; differential frost heave; orthogonal design; residual strength; fitting formula

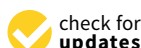



## 1. Introduction

As we all know, pipeline engineering is the most economical transportation for oil and gas energy. It is called the lifeline project of oil and gas transportation. The pipeline engineering inevitably goes through permafrost regions and seasonal permafrost regions. Due to the severe cold climate along the pipeline in permafrost regions, the physical state of frozen soil occurs periodic and irreversible frost and thaw under the effect of the cyclical change of the surface temperature and the temperature of the medium in the pipeline [1,2], the differential frost heave of the soil often occurs. According to the survey, differential frost heave is one of the main pipeline–soil diseases in frozen soil areas. In addition, due to the complex environment in which the frozen soil pipeline is located, there will be many defects in the frozen soil pipeline, which have a decisive impact on the safety and economy of the frozen soil pipeline. Although anticorrosive materials have a certain anticorrosive effect [3], many buried pipelines in service are corroded with the increase of service time. Among the various defects of pipelines in frozen soil areas, internal corrosion defects are an

important factor threatening the safe operation of pipelines. When the pipeline is corroded by the corrosive medium in the pipeline, the wall thickness of the local pipeline will be thinned and the carrying capacity will decrease. In severe cases, corrosion perforation will occur [4–6].

At present, scholars from various countries mainly use finite element simulation analysis and experimental simulation to analyze and evaluate the stress status of pipelines and the residual strength of pipelines with corrosion defects in frozen soil areas [7,8]. Gao et al. [9] used a combination of finite element simulation analysis and experimental simulation to show that the method based on the full-scale blasting experiment of the sample pipeline may overestimate or underestimate the strength of the residual corrosion pipeline, and then a method based on the von Mises criterion to predict the residual strength of the pipeline with long corrosion defects was proposed. Chen et al. [10] studied the residual strength of high-strength pipelines with a single corrosion defect and multiple corrosion defects by using the nonlinear finite element analysis method. The failure pressure predicted by the proposed solution was compared with experimental results and various evaluation methods available in the literature, which proved its accuracy and versatility. Arumugam et al. [11] used the finite element method to predict the residual strength of externally corroded pipes with ring grooves under internal pressure. The failure trend of annular groove corrosion was determined, and the equation for predicting pipeline failure pressure was established. Li et al. [12] used ADINA finite element software to establish a three-dimensional model and analyzed the mechanical properties of buried pipelines with internal corrosion defects at the midpoint of the frost-heave section. The results showed that the corrosion depth had a greater effect on the stress–strain of the pipe than the corrosion length. There are mainly ASME B31G-2012 standards, DNV RP-F101 standards and PCORRC methods for evaluating the residual strength of pipelines with a single corrosion defect [13–15]. Although the application code can be used to calculate the residual strength of the pipeline under various working conditions, the workload is relatively cumbersome, and the stress state of the frozen soil pipeline can not be seen intuitively. In order to analyze the residual strength of the pipeline with corrosion defects in the frozen soil region more reasonably and effectively, the thermo-mechanical coupling analysis of buried non-corroded pipelines in seasonally frozen soil regions was carried out by using ABAQUS finite element software to determine the location of the maximum Mises equivalent stress on the inner wall of buried pipelines in this paper [16,17]. On this basis, we continue to explore the mechanical properties of buried pipelines with internal corrosion, and the effects of different corrosion depth, corrosion length, corrosion width and other factors on the residual strength of the pipeline. The method in this paper can be used to analyze the stress changes and residual strength of buried pipelines with internal corrosion defects in seasonally frozen soil regions. The methods and conclusions can provide valuable references for simulation real-time monitoring and safety analysis in the pipeline operation stage.

## 2. Establishment of Three-Dimensional Pipeline-Soil Thermo-Mechanical Coupling Model

### 2.1. Pipeline-Soil Model Parameters

#### 2.1.1. SOIL Material Parameters

According to the geological exploration data of the frozen soil in Northeast China [18–24], the modelled depth was selected to be 20 m below the natural ground, the modelled length was 100 m, and the modelled width was 40 m. The soil of buried pipeline was divided into non-frost heaving section (10 m) + transition section (20 m) + frost heaving section (40 m) + transition section (20 m) + non-frost heaving section (10 m) = 100 m. The simplified soil layers from top to bottom were silty clay layer with a thickness of 3 m, fine sand and gravel layer with a thickness of 7 m and weathered bedrock residue layer with a thickness of 10 m. The diameter of the pipeline was 0.762 m, the wall thickness was 0.0175 m, the thickness of the insulation layer was 8 cm, and the buried depth was 1.6 m.

The pipeline–soil system was a spatial model. Taking into account the symmetry of the model, the finite element calculation model uses 1/4 (1/2 in both vertical and horizontal directions) for analysis. The two-dimensional plane model of the pipeline–soil model is shown in Figure 1.

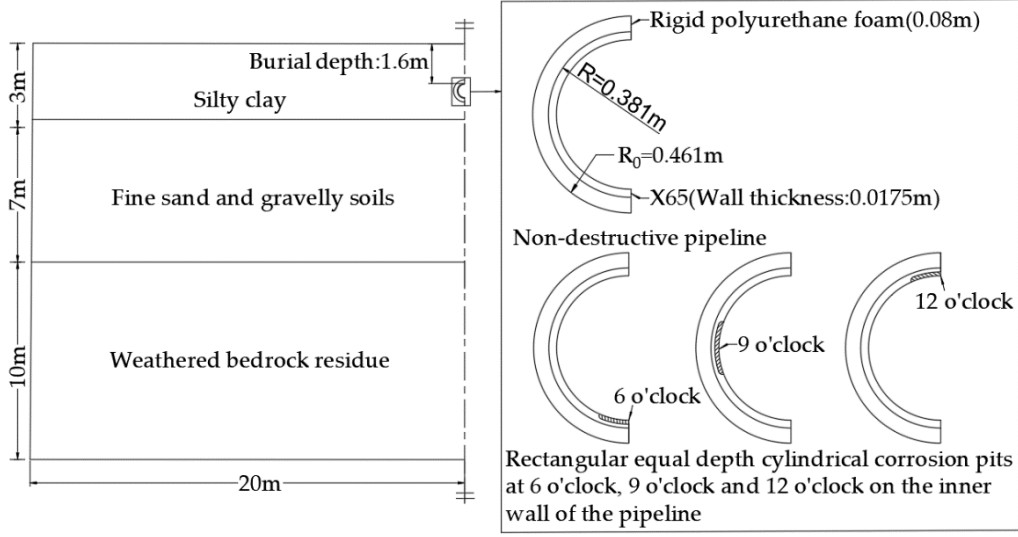

**Figure 1.** Two-dimensional plane model of the pipeline–soil model without damage and with internal corrosion defects.

The soil thermal parameters and moisture content are shown in Table 1 [12], and the soil mechanical parameters of each layer in the frozen soil area are shown in Table 2 [25,26]. Since the thermal expansion and contraction of the soil has no obvious variation, only the frost heave caused by the frozen volume expansion of the water in the soil needs to be considered in the calculation. The volume expansion of water after freezing is about 9%, and the linear expansion coefficient $\alpha$ should be introduced in the calculation [27], and the value is obtained according to the following Formula (1):

$$\alpha = -\frac{0.09\omega}{3\Delta T} \tag{1}$$

where $\omega$ is the natural moisture content (%); $\Delta T$ is the temperature increment (°C).

**Table 1.** Thermal parameters and moisture content of soil.

| | Temperature/°C | | | | | | | | |
|---|---|---|---|---|---|---|---|---|---|
| | −30 | −10 | −5 | −2 | −1 | −0.5 | 0 | 15 | 30 |
| **Silty Clay ($\omega$ = 29.82%)** | | | | | | | | | |
| Thermal conductivity λ (W/m·°C) | 1.21 | 1.21 | 1.21 | 1.21 | 1.21 | 1.21 | 0.84 | 0.84 | 0.84 |
| Specific heat C (J/kg·°C) | 1490 | 1490 | 1490 | 1490 | 1490 | 1490 | 1880 | 1880 | 1880 |
| Soil expansion coefficient $\alpha$ ($10^{-4}$) | 2.982 | 8.946 | 17.890 | 44.730 | 89.460 | 178.9 | 0 | 0 | 0 |
| **Fine Sand and Gravel ($\omega$ = 15%)** | | | | | | | | | |
| Thermal conductivity λ (W/m·°C) | 1.04 | 1.04 | 1.04 | 1.04 | 1.04 | 1.04 | 1.04 | 1.04 | 1.04 |
| Specific heat C (J/kg·°C) | 2540 | 2540 | 2540 | 2540 | 2540 | 2540 | 3350 | 3350 | 3350 |
| Soil expansion coefficient ($10^{-4}$) | 1.500 | 4.500 | 9.000 | 22.500 | 45.00 | 90.00 | 0 | 0 | 0 |

**Table 1.** *Cont.*

| | Temperature/°C | | | | | | | | |
|---|---|---|---|---|---|---|---|---|---|
| | **−30** | **−10** | **−5** | **−2** | **−1** | **−0.5** | **0** | **15** | **30** |
| **Weathered Bedrock Residue ($\omega$ = 2.59%)** | | | | | | | | | |
| Thermal conductivity $\lambda$ (W/m·°C) | 2.12 | 2.12 | 2.12 | 2.12 | 2.12 | 2.12 | 1.42 | 1.42 | 1.42 |
| Specific heat C (J/kg·°C) | 1500 | 1500 | 1500 | 1500 | 1500 | 1500 | 1900 | 1900 | 1900 |
| Soil expansion coefficient $\alpha$ ($10^{-4}$) | 0.259 | 0.777 | 1.554 | 3.885 | 7.770 | 15.54 | 0 | 0 | 0 |

**Table 2.** Soil mechanical parameters.

| | Temperature/°C | | | | | |
|---|---|---|---|---|---|---|
| | **−20** | **−10** | **−5** | **−2** | **0** | **20** |
| **Silty Clay** | | | | | | |
| Density $\rho$ (kg/m$^3$) | 1920 | 1920 | 1920 | 1920 | 1920 | 1920 |
| Poisson's ratio $\nu$ | 0.32 | 0.32 | 0.32 | 0.32 | 0.35 | 0.35 |
| Internal friction angle $\varphi$ (°) | 26 | 26 | 26 | 26 | 24 | 24 |
| Cohesion c (MPa) | 0.6 | 0.6 | 0.6 | 0.57 | 0.15 | 0.15 |
| Elastic modulus E (MPa) | 200 | 100 | 50 | 23.4 | 6 | 6 |
| **Fine Sand and Gravel** | | | | | | |
| Density $\rho$ (kg/m$^3$) | 1834 | 1834 | 1834 | 1834 | 1834 | 1834 |
| Poisson's ratio $\nu$ | 0.15 | 0.15 | 0.15 | 0.15 | 0.2 | 0.2 |
| Internal friction angle $\varphi$ (°) | 20 | 20 | 20 | 20 | 18 | 18 |
| Cohesion c (MPa) | 1.3 | 1.3 | 1.3 | 1.3 | 0.1 | 0.2 |
| Elastic modulus E (MPa) | 500 | 300 | 100 | 70 | 3 | 3 |
| **Weathered bedrock residue** | | | | | | |
| Density $\rho$ (kg/m$^3$) | 2330 | 2330 | 2330 | 2330 | 2330 | 2330 |
| Poisson's ratio $\nu$ | 0.3 | 0.3 | 0.3 | 0.3 | 0.35 | 0.35 |
| Internal friction angle $\varphi$ (°) | 42 | 42 | 42 | 42 | 33.32 | 33.32 |
| Cohesion c (MPa) | 25.2 | 25.2 | 25.2 | 25.2 | 12.6 | 12.6 |
| Elastic modulus E (MPa) | 1400 | 1400 | 1400 | 1400 | 700 | 700 |

2.1.2. Pipe Material Parameters

According to the working conditions, both pipeline and thermal insulation materials are within the standard operating temperature range. Compared with the sensitivity of soil to temperature, the small changes in the temperature-dependent material properties of pipeline and thermal insulation materials can be ignored. The pipe material is X65 steel, the elastic modulus is $2.06 \times 10^5$ MPa, the Poisson's ratio is 0.3, the density is 7850 kg/m$^3$, the thermal conductivity is 60.5 w/(m·°C), the specific heat capacity is 434 J/(kg·°C), the yield strength is 448 MPa, and the tensile strength is 672 MPa. The thermal insulation material is rigid polyurethane foam material, the elastic modulus is 200 MPa, the Poisson's ratio is 0.39, the density is 60 kg/m$^3$, the thermal conductivity is 0.047 w/(m·°C), the specific heat capacity is 1200 J/(kg·°C).

Due to the differential frost heaving of soil, the physical and mechanical characteristics of X65 pipeline material show nonlinear performance (material nonlinearity) and the displacement of the pipeline structure system makes the mechanics of the system change significantly (geometric nonlinearity). Therefore, the Ramberg–Osgood model selected for the constitutive relationship of X65 pipeline in this paper can fully consider the influence of soil on the elastoplastic properties and large deformation of pipeline [28,29]. Furthermore, the true stress–strain curve of the material was calculated as shown in Figure 2.

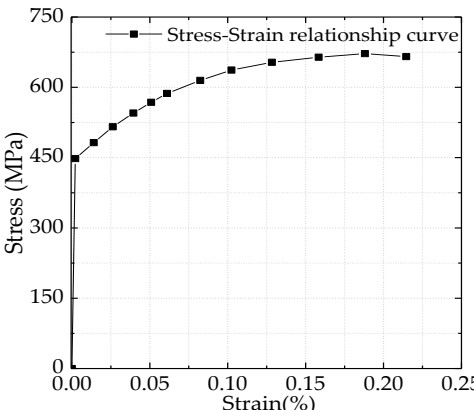

**Figure 2.** The Ramberg–Osgood model curve of X65 pipeline.

### 2.2. Establishment of the Pipeline-Soil Solid Model

The pipeline–soil thermo-mechanical coupling model in this paper used sequential coupling analysis (that is, from the initial temperature field to the cyclic temperature field and then to the stress field). The Mohr–Coulomb elastic–plastic constitutive model was adopted for soil, as shown in Formulas (2) and (3). The failure envelope of Mohr–Coulomb constitutive model combines the shear failure criterion and tensile failure criterion. As shown in Figure 3a the elements of the soil within the range of 5 m near the junction of the soil transition section and the frost heaving section and the soil near the axial direction of the pipeline should be grid refined, with a total of 28,431 elements. In addition, the insulation layer and the non-corroded pipeline were also divided by hexahedron with 2000 and 8000 elements.

$$\sigma_1 - \sigma_3 N_\varphi - 2c\sqrt{N_\varphi} = 0 \tag{2}$$

$$N_\varphi = \tan^2\left(\frac{\pi}{4} + \frac{\varphi}{2}\right) \tag{3}$$

where $\sigma_1$ and $\sigma_3$ are first principal stress and third principal stress, respectively; c and $\varphi$ are the cohesion and internal friction angle of frozen soil, respectively.

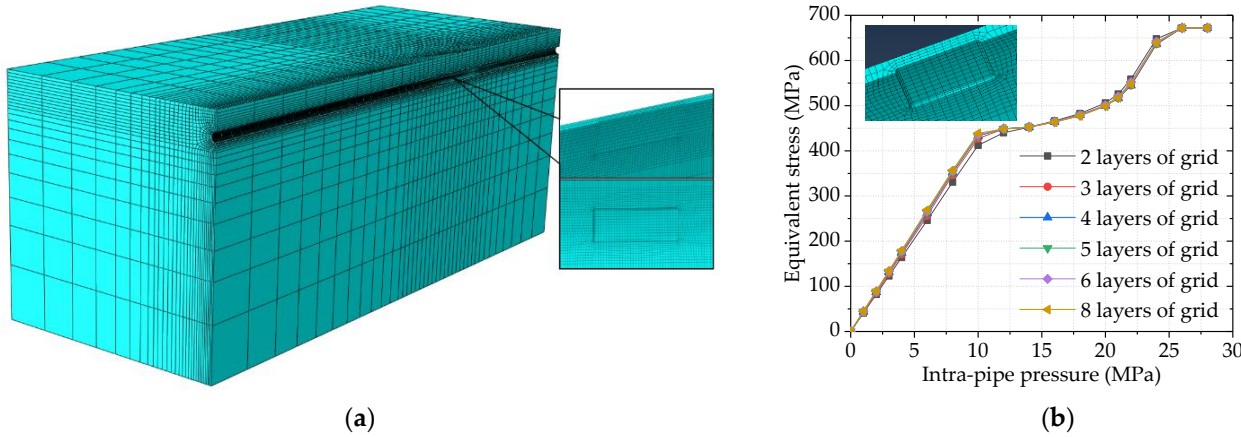

(**a**)  (**b**)

**Figure 3.** (**a**) Three-dimensional pipe-soil thermal-mechanical coupling model for non-corroded and internally corroded defective pipelines; (**b**) Trend of equivalent stresses with a different number of grid layers.

In order to improve the calculation accuracy and efficiency of the study area, the non-corrosion pit area of the corroded pipeline was roughly meshed along the axis of the pipeline, and the corrosion pit area was densified by fine grid size; in the direction along the wall thickness of the corrosion area, when the corrosion depth was 15% to 80%, the appropriate grid division method was obtained by comparing the variation curve of Mises equivalent stress on the inner wall of the pipeline with internal pressure when dividing

grids with different layers. For example, when the corrosion depth was 25%, the Mises equivalent stress curves obtained by the simulation analysis of the hexahedral mesh with different numbers of layers were compared, as shown in Figure 3b. It can be found that four layers in the corrosion area and two layers in other areas will be able to obtain more accurate results. At this time, there were 24,716 hexahedral elements in corroded pipelines.

The boundary conditions are shown in Figure 4. Symmetric constraints were applied to the Z = −50 m and X = 0 sections of the pipeline–soil coupling model. Setting a single displacement constraint on Z = 0, Y = −20 m and X = −20 m sections; the ground at Y = 0 is free constraint. The contact surface between the insulation layer and the pipeline was bound. The contact property between the insulation layer and the soil was set as surface-surface contact, the vertical direction was set as hard contact, when the horizontal friction coefficient was 0.3, the real tangential behavior between pipeline and soil can be well reflected. When the temperature field and stress field were analyzed, the hexahedral heat transfer DC3D8 element and the three-dimensional stress C3D8R element were selected respectively. In this paper, a three-dimensional pipeline–soil thermo-mechanical coupling model of a 1/4 symmetrical non-corroded pipeline was established to determine the location of the maximum Mises equivalent stress on the inner wall of the buried pipeline. Furthermore, on this basis, a three-dimensional pipeline–soil thermo-mechanical coupling model of a 1/4 symmetrical pipeline with internal corrosion defects was established.

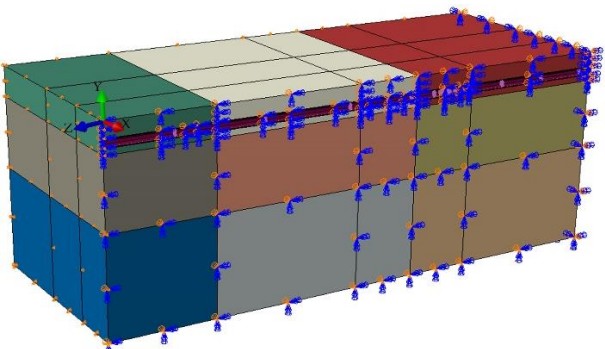

**Figure 4.** Three-dimensional pipeline–soil thermal-mechanical coupling model and its boundary conditions.

### 2.3. Verification and Analysis of the Temperature Field

The temperature initial conditions for the thermally coupled model in this paper: the coupled temperature field, averaged over the next 50 years, was used as the initial temperature field of the soil and this was carried over into the cyclic temperature field. The thermostatic boundary at the bottom of the model (at 20 m) was taken to be −1 °C. Temperature boundary conditions on the sides of the model were set to insulated boundaries. The temperature boundary conditions for the upper boundary of the model were determined according to the temperature variation function Equation (4) for the northeast region of China [30,31].

$$T(t) = -3.5 + \frac{0.048}{8760}t_0 + A_0 \sin\left(\frac{2\pi}{8760}t_0 + \varphi\right) \tag{4}$$

where $t_0 = 3600t$(s) (where $t$ is the pipeline operating time h); $A_0$ is the temperature change amplitude; and $\varphi$ is the initial phase angle.

When $A_0 = 20$ °C and $\varphi = \pi/2$, the temperature profile within the second decade was calculated as shown in Figure 5, with a maximum temperature of 17.46 °C and a minimum temperature of −22.563 °C.

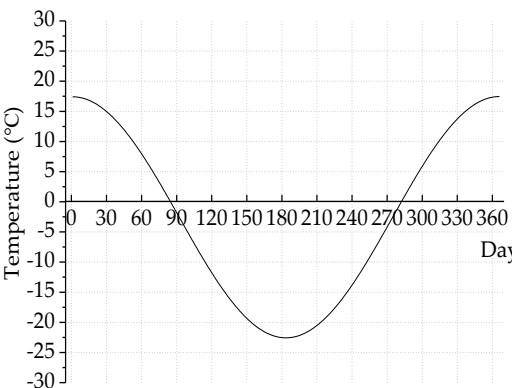

**Figure 5.** Curve of temperature change in the 20th year.

In order to ensure the accuracy of the stress field in the pipeline–soil thermo-mechanical coupling model, we first verified the temperature field of the coupling model. In the numerical simulation, the average transport temperature of the medium in the tube was adopted from the measured oil temperature of 3 °C. The results were compared with field monitoring of the temperature at section AB039−100 in February 2012, at a location 11.5 m from the center of the pipe [32,33]. The temperature field cloud diagram was calculated by the finite element analysis as shown in Figure 6. The comparison with the calculated temperature field curve and the actual temperature field curve was shown in Figure 7. The results show that the calculated results were close to the measured data, so the numerical calculations can simulate the actual heat transfer situation in the corresponding environment.

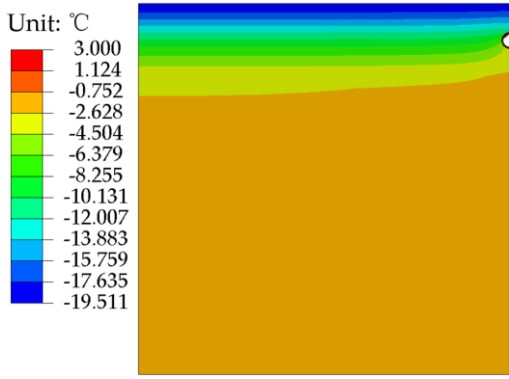

**Figure 6.** Temperature field distribution clouds for February for the pipe soil thermal model.

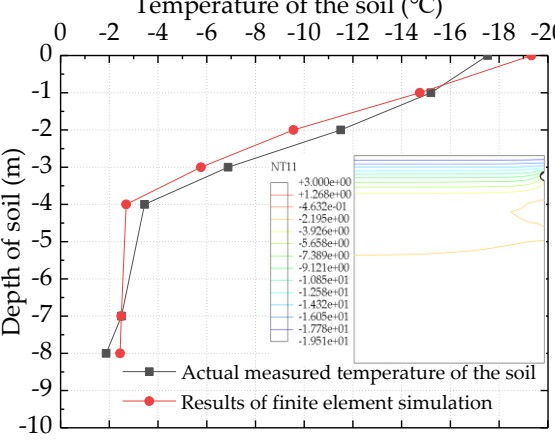

**Figure 7.** Curve of simulated soil temperature and measured temperature.

### 3. Mechanical Analysis of Inner Wall of Buried Non-Corroded Pipeline in Seasonal Frozen Soil Region

*3.1. The Effect of Differential Frost Heave of Soils on the Deformation of Buried Pipelines*

The diameter, wall thickness and burial depth of the buried non-corroded pipe (X65) were known to be 0.762 m, 0.0175 m and 1.6 m, respectively. The specific boundary conditions of the model were described in Section 2.2. When the internal pressure of the pipe is 10.0 MPa and the temperature of the medium in the pipe is 3 °C. Combined with the two-dimensional plane diagram of the synergistic deformation of pipeline–soil in Figure 8. The change law of vertical deformation and Mises equivalent stress of buried non-corroded pipelines in the seasonally frozen-ground region with temperature change (within the 20th year) is as follows.

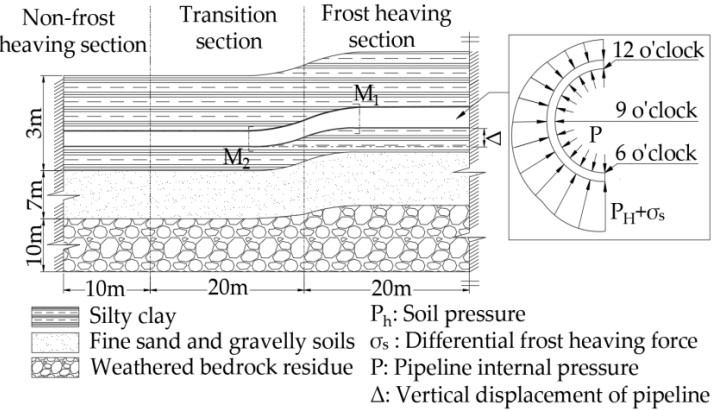

**Figure 8.** The simplified figure of differential frost heaving deformation of soil and buckling deformation of pipeline.

1.  According to the Vertical Displacement Curve as shown in Figure 9, the average vertical displacement of buried pipelines increases with decreasing temperature when at 90 D (October) to 240 D (March). The maximum vertical displacement occurred in the frost heave section of 180 D (January), which was 0.033 m. At this time, the vertical displacement deformation of the top, bottom and sides (9 o'clock direction) of the inner wall of the buried pipeline is shown in Figure 10, which reflects the deformation form of the buried pipeline under the differential frost heaving of the soil when the temperature is the lowest.

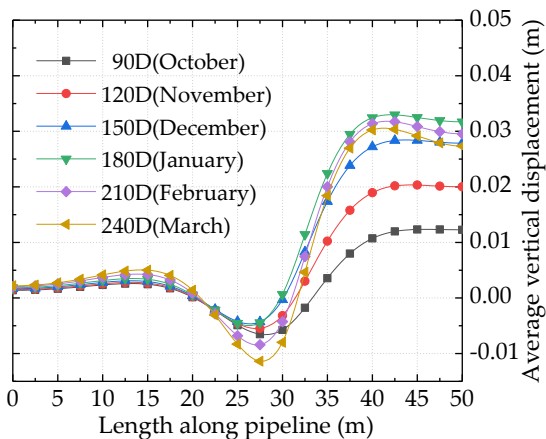

**Figure 9.** Vertical displacement of the pipeline as a function of temperature.

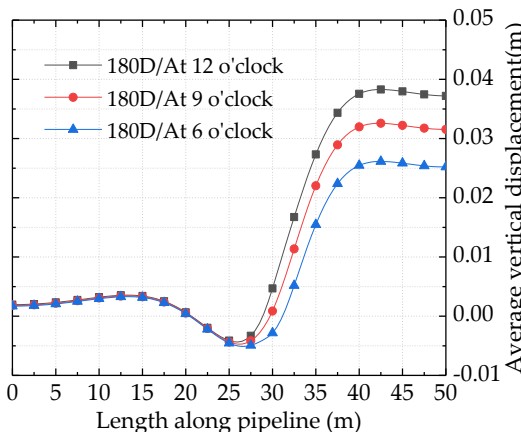

**Figure 10.** Vertical displacement of the pipeline at the lowest temperature (180 D).

2.  The effect of subzero temperature from 120 D (November) to 210 D (February) on Mises equivalent stress is shown in Figure 11. When the temperature is lowest at 180 D (January), Mises equivalent stress is maximum. At this time, the Mises equivalent stress peak points, 241.6 MPa and 323.8 MPa, occur at the top (12 o'clock) of the inner wall of the pipeline approximately 1.5 m to the left of the junction of the soil transition section and the frost heave section and at the side (9 o'clock) of the inner wall of the pipeline approximately 2.5 m to the right of this junction.

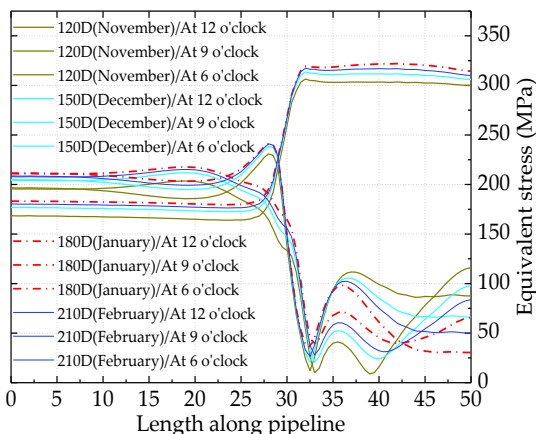

**Figure 11.** The law of the equivalent stress of the inner wall of pipeline changing with temperature.

3.  Due to the differential frost heaving of soil, the buried pipeline generates buckling deformation near the junction of the transition section and the frost heaving section. The offset along the pipeline length reaches the maximum in the buckling section, but it is much smaller than the vertical offset of the pipeline. In addition, due to the existence of circumferential extrusion force and axial friction force of soil on the pipeline, the offset along the length direction of the pipeline gradually decreases from the buckling section to both sides and finally tends to zero. As shown in Figure 8, the non-uniform tension between the cross-section $M_1$ and cross-section $M_2$ of the pipeline leads to the non-linear shear stress on the inner wall of the pipeline in the buckling section.

### 3.2. Effects of Different Internal Pressures on the Mechanical Properties of Buried Pipelines

In summary, when the temperature of 180 D (January) was the lowest, the buried pipeline reached an extreme buckling state near the junction of the soil transition section and the frost heave section. The Mises equivalent stress curves at the top (12 o'clock), bottom (6 o'clock) and sides (9 o'clock) of the inner wall of buried non-corroded pipeline

were studied at different internal pressures when the temperature of the medium of the pipeline is 3 °C.

From Figure 12, it can be seen that when the internal pressure of the pipeline has had below 22.0 MPa, the maximum Mises equivalent stress appears on the side(9 o'clock) of the inner wall of the pipeline 2.5 m on the right side of the junction of the soil transition section and the frost heave section. By contrast, when the internal pressure of the pipeline has had above 22.0 MPa, the maximum Mises equivalent stress occurs at the top (12 o'clock) of the inner wall of the pipeline 1.5 m on the left side at the junction of the soil transition section and the frost heaving section.

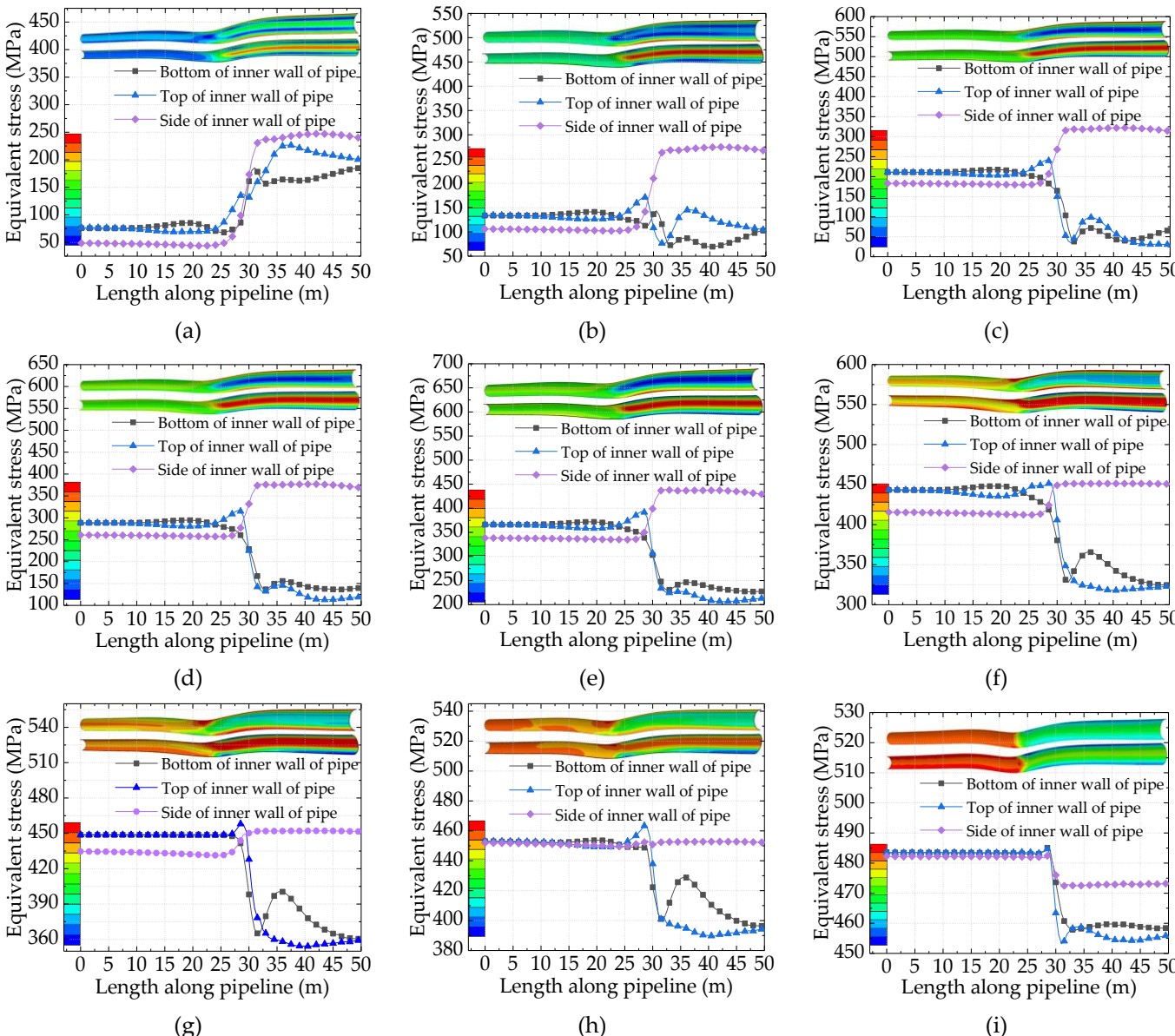

**Figure 12.** Mises equivalent stress clouds and curves for the inner wall of the pipe at different internal pressures. (**a**) Internal pressure 3.0 MPa, (**b**) Internal pressure 6.0 MPa, (**c**) Internal pressure 10.0 MPa, (**d**) Internal pressure 14.0 MPa, (**e**) Internal pressure 18.0 MPa, (**f**) Internal pressure 22.0 MPa, (**g**) Internal pressure 23.0 MPa, (**h**) Internal pressure 24.0 MPa, (**i**) Internal pressure 26.0 MPa.

## 4. Study on Residual Strength of Buried Pipelines with Single Internal Corrosion Defect in Seasonal Frozen Soil Region

According to the location of the maximum Mises equivalent stress on the inner wall of the buried non-corroded pipeline determined in Section 3.2, combined with the orthogonal design method, the three-dimensional pipeline–soil thermo-mechanical coupling model of 1/4 pipelines with internal corrosion defects was established. The effect of different corrosion depths, lengths and widths on the residual strength of buried pipelines with internal corrosion defects in seasonally frozen soil regions and the mechanical characteristics of the corrosion area on the inner wall of pipelines was analyzed. Based on the plastic failure criterion theory, this paper considered the transportation performance and the risk of pipeline leakage after pipeline deformation. In total, 80% of the ultimate tensile strength $\sigma_u$ (672 MPa) was selected as the reference stress [34]. The failure conditions are shown in Formulas (5), that is, the pipeline fails when the Mises equivalent stress in the corrosion defect area reaches the reference stress, and the failure critical value is 537.6 MPa.

$$\sigma_v = \left\{ \frac{1}{2}\left[(\sigma_1 - \sigma_2)^2 + (\sigma_2 - \sigma_3)^2 + (\sigma_3 - \sigma_1)^2\right] \right\}^{\frac{1}{2}} < [\sigma] = 0.8\sigma_u \tag{5}$$

where $\sigma_v$ denotes the Mises equivalent stress, MPa; $[\sigma]$ denotes the reference stress, MPa.

### 4.1. Orthogonal Design Method

Orthogonal design is a design method to study multi-factor and multi-level. According to the orthogonality, some representative points are selected from the comprehensive simulation test to carry out the simulation test, which can achieve the equivalent results with a large number of comprehensive simulation tests with the least number of simulation tests. So according to the orthogonal design method, the internal corrosion defect parameters: corrosion length, corrosion depth, and corrosion width were taken as three factors (A, B, and C). Five levels were selected for each factor, and the orthogonal design combination was carried out through the L25 orthogonal table. The specific working conditions are shown in Table 3.

**Table 3.** Orthogonal design table (mm).

|  | A<br>Corrosion Depth | B<br>Corrosion Length | C<br>Corrosion Width |
|---|---|---|---|
| 1 | 1 (2.625) | 1 (100) | 1 (50) |
| 2 | 1 (2.625) | 2 (200) | 2 (100) |
| 3 | 1 (2.625) | 3 (300) | 3 (200) |
| 4 | 1 (2.625) | 4 (600) | 4 (300) |
| 5 | 1 (2.625) | 5 (900) | 5 (600) |
| 6 | 2 (4.375) | 1 (100) | 3 (200) |
| 7 | 2 (4.375) | 2 (200) | 4 (300) |
| 8 | 2 (4.375) | 3 (300) | 5 (600) |
| 9 | 2 (4.375) | 4 (600) | 1 (50) |
| 10 | 2 (4.375) | 5 (900) | 2 (100) |
| 11 | 3 (8.750) | 1 (100) | 5 (600) |
| 12 | 3 (8.750) | 2 (200) | 1 (50) |
| 13 | 3 (8.750) | 3 (300) | 2 (100) |
| 14 | 3 (8.750) | 4 (600) | 3 (200) |
| 15 | 3 (8.750) | 5 (900) | 4 (300) |
| 16 | 4 (13.125) | 1 (100) | 2 (100) |
| 17 | 4 (13.125) | 2 (200) | 3 (200) |
| 18 | 4 (13.125) | 3 (300) | 4 (300) |
| 19 | 4 (13.125) | 4 (600) | 5 (600) |
| 20 | 4 (13.125) | 5 (900) | 1 (50) |

**Table 3.** *Cont.*

|  | A<br>Corrosion Depth | B<br>Corrosion Length | C<br>Corrosion Width |
|---|---|---|---|
| 21 | 5 (14.000) | 1 (100) | 4 (300) |
| 22 | 5 (14.000) | 2 (200) | 5 (600) |
| 23 | 5 (14.000) | 3 (300) | 1 (50) |
| 24 | 5 (14.000) | 4 (600) | 2 (100) |
| 25 | 5 (14.000) | 5 (900) | 3 (200) |

*4.2. The Varying Regularity of Mises Equivalent Stress in the Corrosion Region of Buried Pipeline under Failure Pressure*

Combined with the analysis of Section 3.2, the residual strength of the buried defect pipeline was determined by comparing the failure pressure of corrosion defects at the top and side of the pipeline inner wall. For example, according to the simulation analysis of No. 11, the failure pressure of the buried pipeline was 23.32 MPa when the internal corrosion defect was located at the side (9 o'clock) of the inner wall of the buried pipeline in the soil frost heaving section. When the internal corrosion defect was located at the top (12 o'clock) of the inner wall of the buried pipeline in the soil transition section, the failure pressure of the buried pipeline was 21.75 MPa, and the relative pressure resistance was reduced by 6.7%, as shown in Figure 13. Therefore, the residual strength of the No. 11 buried defect pipeline was 21.75 MPa. It was also demonstrated that when the internal pressure of the pipeline exceeded 22.0 MPa, the dangerous position of the inner wall of the buried pipeline would obliquely transit from the side (9 o'clock) of the inner wall of the pipeline of the 2.5 m right side at the junction of the soil transition section and the frost heaving section to the top (12 o'clock) of the inner wall of the pipeline of the 1.5 m left side at this junction.

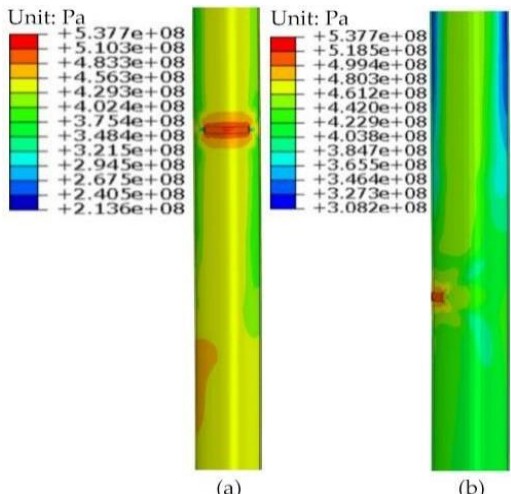

**Figure 13.** Mises equivalent stress comparison diagram of No. 11 when internal corrosion defects are located at different locations: (**a**) Failure pressure is 23.32 MPa; (**b**) Failure pressure is 21.75 MPa.

When the corrosion area was located at the top (12 o'clock) of the inner wall of the buckling section of the pipeline, the Mises equivalent stress within the corrosion area of the inner wall of the pipeline under the failure pressure was generally funnel-shaped. For example, the variation of equivalent stress along the circumferential and axial direction in the corrosion area of No. 4 and No. 5 models are shown in Figure 14.

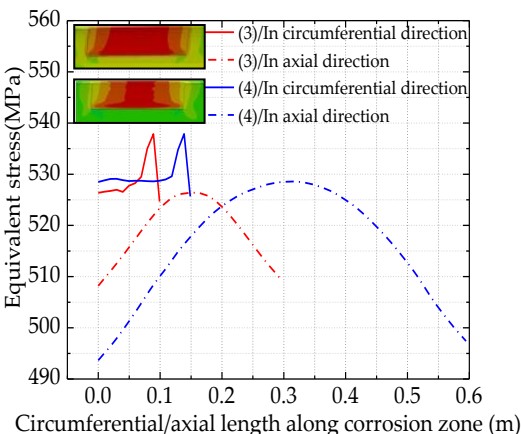

**Figure 14.** Mises equivalent stress variation curves in the corrosion zone.

When the corrosion zone was located on the side (9 o'clock) of the inner wall of the pipeline, near the inner corrosion area, the stress gradient along the circumferential direction of the pipeline was larger than that along the axial direction of the pipeline. It can be seen from the overall distribution form that Mises equivalent stress shows antisymmetric distribution along the center of the corrosion zone. In addition, when the corrosion length ≥ 600 mm, under the failure pressure, there is a sudden change in the equivalent force for every 150 mm interval along the pipe axis, so that the equivalent force outside the corrosion zone along the axial side of the pipeline shows an uneven arc distribution with continuous equal chord length, moreover, there is a certain stress concentration at the edge, as shown in Figure 15.

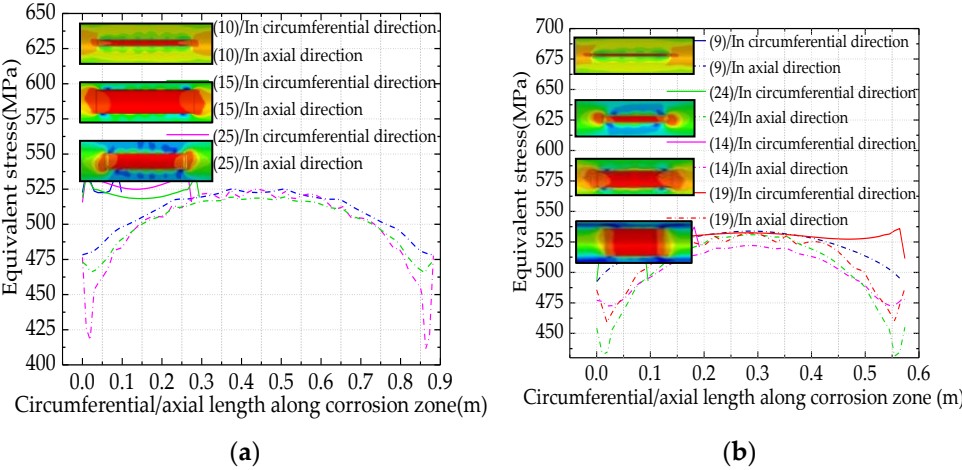

**Figure 15.** The Mises equivalent stress curve when the corrosion zone is located on the side (9 o'clock) of the inner wall of the pipeline: (**a**) The corrosion length is greater than 600 mm and the corrosion width is greater than 50 mm; (**b**) The corrosion length is greater than 900 mm and the corrosion width is greater than 50 mm.

### 4.3. One-Way Analysis of Extreme Variance for Orthogonal Design Method Result Data

According to the corrosion parameters in Table 3 of orthogonal design and simulated by ABAQUS finite element software, the residual strengths of 25 pipelines were calculated, and the simulated data were collated and analyzed with the mathematical statistics method, as shown in Table 4.

**Table 4.** Analysis of extreme variance of the resultant data.

| Results of the Simulation | | | | Analysis of Results Data | | |
| --- | --- | --- | --- | --- | --- | --- |
| | | | | **A** | **B** | **C** |
| **Number** | **Residual Strength** | **Number** | **Residual Strength** | **Corrosion Depth** | **Corrosion Length** | **Corrosion Width** |
| 1 | 25.69 | 14 | 16.10 | | 124.200 | 110.710 | 85.110 |
| 2 | 25.43 | 15 | 15.43 | | 114.180 | 93.170 | 94.210 |
| 3 | 25.09 | 16 | 20.93 | T | 92.320 | 87.110 | 86.820 |
| 4 | 24.08 | 17 | 13.43 | | 61.910 | 78.160 | 91.160 |
| 5 | 23.91 | 18 | 11.22 | | 52.070 | 75.530 | 87.380 |
| 6 | 25.22 | 19 | 8.81 | | | | |
| 7 | 23.31 | 20 | 7.52 | | 24.840 | 22.142 | 17.022 |
| 8 | 22.19 | 21 | 17.12 | | 22.836 | 18.634 | 18.842 |
| 9 | 21.77 | 22 | 10.72 | t | 18.464 | 17.422 | 17.364 |
| 10 | 21.69 | 23 | 9.85 | | 12.382 | 15.632 | 18.232 |
| 11 | 21.75 | 24 | 7.40 | | 10.414 | 15.106 | 17.476 |
| 12 | 20.28 | 25 | 6.98 | | | | |
| 13 | 18.76 | | | R | 14.426 | 7.036 | 1.820 |

In Table 4, A, B and C denote corrosion depth, corrosion length and corrosion width (mm) respectively; T denotes the sum of the data of each level factor for each factor; t denotes the average value of the data of each level factor; R denotes the range of the average value of the data of each level factor, and its value indicates the degree of influence of the factor on the residual strength (MPa) of the defect pipeline.

According to Table 4 analysis:

1. When the corrosion length was constant, the residual strength curve became steeper and steeper as the depth–thickness ratio of the corrosion defects in buried pipelines increased from 15% to 80%, as shown in Figure 16a; the relative decrease rate of residual strength of buried corrosion defect pipeline gradually increased, as shown in Table 5. In addition, the average residual strength of the pipeline changed from 24.840 MPa to 10.414 MPa, which was reduced by 14.426 MPa. The numerical fluctuation was about 2 times the corrosion length and 8 times the corrosion width, indicating that the corrosion depth was the main factor affecting the residual strength of the pipeline.

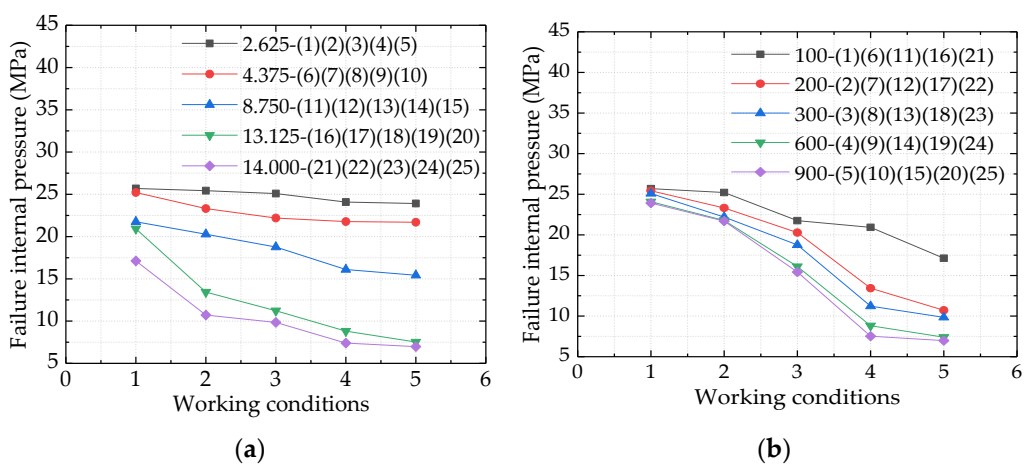

**Figure 16.** Residual strength curve of buried corroded pipeline: (**a**) Different corrosion depths; (**b**) Different corrosion lengths.

**Table 5.** Effect of increasing corrosion depth on the residual strength of the pipeline when the corrosion length is constant.

| Number | Relative Reduction Rate of Residual Strength of Pipeline | | | |
|---|---|---|---|---|
| (1)(6)(11)(16)(21) | 1.8% | 15.3% | 18.5% | 33.4% |
| (2)(7)(12)(17)(22) | 8.3% | 20.3% | 47.2% | 57.8% |
| (3)(8)(13)(18)(23) | 11.6% | 25.2% | 55.3% | 60.7% |
| (4)(9)(14)(19)(24) | 9.6% | 33.1% | 63.4% | 69.3% |
| (5)(10)(15)(20)(25) | 9.3% | 35.5% | 68.5% | 70.8% |

1. When the corrosion depth was constant, the corrosion length increased from 100 mm to 600 mm, as shown in Figure 16b, the residual strength curve also becomes steeper and steeper. The residual strength of the buried pipeline with internal corrosion defects decreased gradually, and the average residual strength of the pipeline changed from 22.142 MPa to 15.632 MPa, which was reduced by 6.51 MPa. However, when the corrosion length increased from 600 mm to 900 mm, the relative reduction rate of residual strength decreased gradually, as shown in Table 6. At this time, the average residual strength of the pipeline changed from 15.632 MPa to 15.106 MPa, only reduced by 0.526 MPa. In conclusion, the corrosion length had a great effect on the residual strength of the pipeline. When the corrosion length exceeded 600 mm, the effect degree would gradually decrease.

**Table 6.** Effect of increasing corrosion length on the residual strength of the pipe when the corrosion depth is constant.

| Number | Relative Reduction Rate of Residual Strength of Pipeline | | | |
|---|---|---|---|---|
| (1)(2)(3)(4)(5) | 1.0% | 2.3% | 6.3% | 6.9% |
| (6)(7)(8)(9)(10) | 7.6% | 12.0% | 13.7% | 14.0% |
| (11)(12)(13)(14)(15) | 6.8% | 13.7% | 26.0% | 29.1% |
| (16)(17)(18)(19)(20) | 35.8% | 46.4% | 57.9% | 64.1% |
| (21)(22)(23)(24)(25) | 37.4% | 42.5% | 56.8% | 59.2% |

1. When the corrosion width increased from 50 mm to 600 mm, as shown in Figure 17, the failure pressure of the defective pipeline varied irregularly from high to low, but which showed a slight overall decreasing trend. It can be seen from Table 4 that the average residual strength of the pipeline decreased from 18.842 MPa to 17.022 MPa, only 1.82 MPa was reduced, and the influence of corrosion width on the residual strength of pipeline was limited. This meant that as the corrosion width increased the effect degree on the residual strength of the defective pipeline was less.

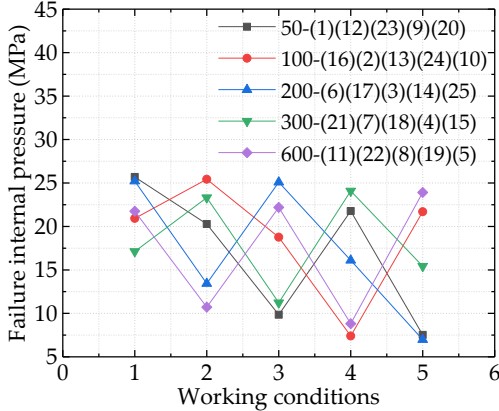

**Figure 17.** Effect of corrosion width on residual strength.

### 4.4. Fitting Formula of Residual Strength of Pipeline with Internal Corrosion Defects and Its Accuracy Verification

Two polynomial model fitting methods were compared by SPSS, as shown in Table 7. When the number of fittings was higher, the $R^2$ value was larger, and the fitting effect was better. In addition, considering the possible interaction between various factors, the second-order polynomial model was selected for linear/nonlinear fitting. The $R^2 = 0.973$ for this model fitting method, meaning that the dependent variable was explained with 97.3% confidence by the independent variable. Moreover, the Durbin–Watson coefficient was around 2.0, and the normal P-P diagram and histogram of the regression standardized residual of the model are shown in Figure 18. It can be seen that the residuals are small and the random errors obey a normal distribution, which also indicates that the fitting formula can accurately predict the residual strength of the pipeline with internal corrosion defects.

**Table 7.** Comparison of two model fitting formula methods.

| Fitting Method | | $R^2$ | Sum of Squares | d-f | Mean Square | F | Sig | Durbin -Watson | Remark |
|---|---|---|---|---|---|---|---|---|---|
| First-order polynomial model | Linear fitting | 0.897 | 909.916 | 3 | 303.305 | 60.697 | <0.0001 | 1.479 | |
| Second-order polynomial model | Linear or non-linear fitting | 0.973 | 987.131 | 10 | 98.713 | 49.848 | <0.0001 | 2.331 | use |

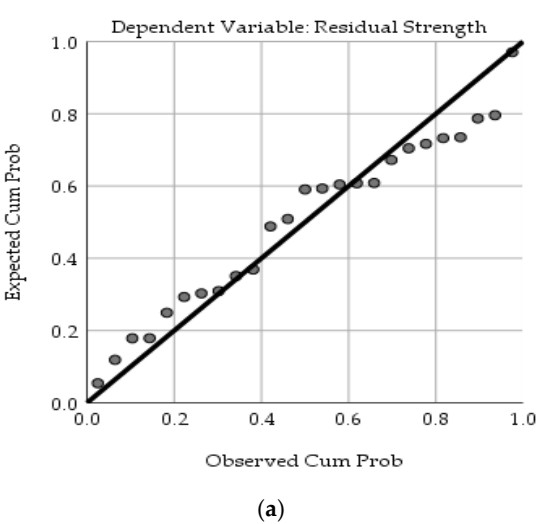
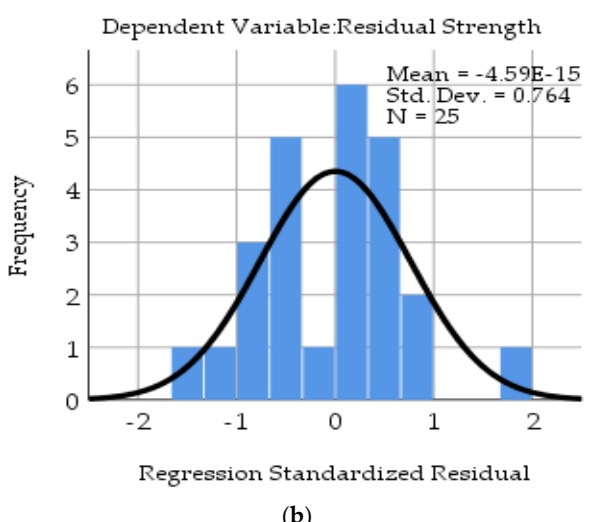

**(a)** **(b)**

**Figure 18.** The normal P-P diagram (**a**) and histogram (**b**) of the regression standardized residual of the mode.

According to Table 7, the linear/nonlinear fitting equation of the quadratic polynomial of the dependent variable Y (residual strength) to the three independent variables (A, B and C) is:

$$Y = \begin{aligned} &28.614 - 0.145A - 0.017B + 0.009\,C - 0.027A^2 + 2.03E^{-5}\,B^2 \\ &-3.92E^{-6}C^2 - 0.001(AB + AC) - 1.26E^{-5}BC + 2.02E^{-6}ABC \end{aligned} \tag{6}$$

where 2.625 mm < A < 14 mm; 100 mm < B < 900 mm; 50 mm < C < 60 mm.

The comparison curve between the finite element simulation value and the predicted value of the fitting formula is shown in Figure 19a. The maximum fitting residual is 2.66 and the minimum is 0.03, indicating that the fitting result is ideal.

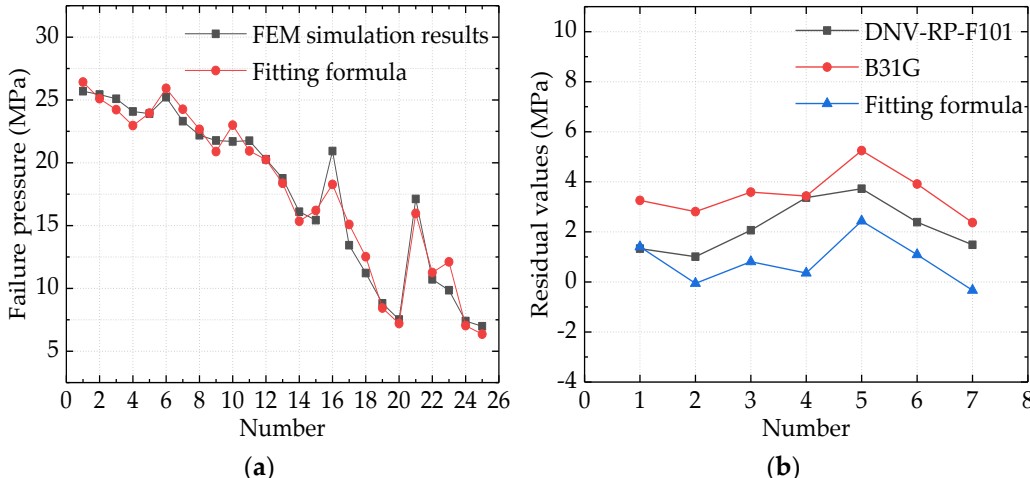

**Figure 19.** (**a**) The comparison of calculation values by finite element model with values of fit formula; (**b**) Comparison curves for residual values.

According to the literature [35,36], the residual strength values of pipelines with corrosion defects predicted by ASME B31G, DNV RP-F101 and the fitting equation were compared with the bursting test data of pipelines with corrosion defects in the relevant literature [37], as shown in Table 8. Combined with Figure 19b, it can be seen that the maximum residual error and minimum residual error of residual strength of the corrosion-defective pipeline predicted by the fitting equation were 2.436 and 0.058, respectively, and the residual sum of squares was 9.984. The predicted results of the fitting formula are better than those of ASME B31G and DNV RP-F101 methods. It can meet the prediction requirements of residual strength of oil and gas transmission pipeline with corrosion defects in seasonally frozen soil regions.

**Table 8.** Verification of the results of the fitted equations with the pipeline bursting data, DNV-RP-F101 and B31G evaluation results.

| Number | Corrosion Length (mm) | Corrosion Width (mm) | Corrosion Depth (mm) | DNV-RP-F101 | B31G | Blast Data | Fitting Formula |
|---|---|---|---|---|---|---|---|
| 1 | 100 | 50 | 8.8 | 22.97 | 21.04 | 24.30 | 22.90 |
| 2 | 200 | 50 | 4.4 | 23.10 | 21.3 | 24.11 | 24.17 |
| 3 | 200 | 50 | 8.8 | 19.69 | 18.17 | 21.76 | 20.95 |
| 4 | 200 | 50 | 13.1 | 13.78 | 13.72 | 17.15 | 16.80 |
| 5 | 200 | 100 | 8.8 | 19.69 | 18.17 | 23.42 | 20.98 |
| 6 | 200 | 200 | 8.8 | 19.69 | 18.17 | 22.08 | 20.99 |
| 7 | 300 | 50 | 8.8 | 17.60 | 16.71 | 19.08 | 19.41 |
| | Maximum error | | | 3.727 | 5.250 | | 2.436 |
| | Minimum error | | | 1.010 | 2.370 | | 0.058 |
| | Residual sum of squares | | | 40.221 | 90.644 | | 9.984 |

## 5. Conclusions

We have considered the complex working conditions of buried pipelines with internal corrosion defects in seasonally frozen soil regions, and a finite element model of a three-dimensional pipeline–soil thermo-mechanical coupling model was established. The temperature field has been verified and the thermo-mechanical coupling analysis has been carried out in this paper. The stress distribution characteristics of the inner wall of the buried pipeline without damage under different pipeline internal pressures were analyzed, and the maximum location of the Mises equivalent stress in the inner wall of the pipeline was determined. On this basis, the equivalent stress distribution characteristics

and residual strength of buried pipeline with internal corrosion defects were studied by orthogonal design method, and the following conclusions were obtained:

The location of the maximum Mises equivalent stress on the inner wall of the non-destructive pipeline is near the junction of the transition section and the frost heave section. When the temperature is the lowest, the Mises equivalent stress is the largest.

1. The residual strength of pipelines with internal corrosion defects in permafrost regions can be evaluated safely and reliably by the orthogonal analysis method. The calculation is simple and convenient for engineering applications.
2. The corrosion depth of the pipeline in the frozen soil area is the main factor affecting the residual strength of the pipeline; as the depth of corrosion defects increases, the residual strength decreases. The corrosion length is the second; but when the corrosion length reaches 600 mm, its effect on the residual strength of the pipeline is no longer significant. The corrosion width has the least effect on the residual strength.
3. Based on the finite element numerical simulation data, a formula for calculating the residual strength of the pipeline with internal corrosion defects in seasonally frozen soil region was obtained by fitting. Compared with the existing corrosion evaluation specifications, the calculation results of the fitting formula obtained according to the stress concentration theory have small errors and uniform error distribution, which can better meet the prediction requirements of failure pressure of oil and gas pipelines with internal corrosion in seasonally frozen soil regions.

**Author Contributions:** Conceptualization, X.L. (Xiaoli Li), X.L. (Xiaoyan Liu), and L.H.; methodology, X.L. (Xiaoli Li), G.C., X.L. (Xiaoyan Liu), J.J., and L.H.; software, G.C.; validation, X.L. (Xiaoli Li), and G.C.; formal analysis, X.L. (Xiaoli Li), G.C.; investigation, X.L. (Xiaoli Li); writing—X.L. (Xiaoli Li), G.C., X.L. (Xiaoyan Liu), J.J., and L.H. All authors have read and agreed to the published version of the manuscript.

**Funding:** This research was partially supported by the National Natural Science Foundation of China (Grant Nos. 51774092, 51534004 and 52076036), and the China Postdoctoral Science Foundation (Grant No. 2016M601399).

**Institutional Review Board Statement:** Not applicable.

**Informed Consent Statement:** Not applicable.

**Data Availability Statement:** Not applicable.

**Acknowledgments:** The authors would like to thank the editor and the reviewers for their contributions. The authors thank Chen Guofei for his help.

**Conflicts of Interest:** The authors declare no conflict of interest.

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
