# Peer review of "Analysis and Evaluation on Residual Strength of Pipelines with Internal Corrosion Defects in Seasonal Frozen Soil Region"

_applsci, doi:10.3390/app112412141_

Round 1
Reviewer 1 Report
Dear authors,
please see some comments and questions in the article.

Author Response
Dear reviewer:
Thank you for your comments concerning our manuscript entitled “Analysis and evaluation on residual strength of pipelines with internal corrosion defects in seasonal frozen soil region” (applsci-1490962).Those comments are all valuable and very helpful for revising and improving our paper, as well as the important guiding significance to our researches. We have studied comments carefully and have made correction which we hope meet with approval. Below we have pasted your comments in blue, our point-by-point response is given in black. Please download the following word file.

Reviewer 2 Report
The present work proposes a prediction formula of residual strength of buried pipelines with internal corrosion defects, obtained by a Statistical Product and Service Solutions fitting. The formula is based on the results of 20 finite element simulations of a buried pipeline subjected to differential soil frost heave. Several sizes of the internal corrosion defect were considered and the prediction results were compared with the experimental data of hydraulic blasting. The prediction error of the fitting formula is small. However, there are some questions and concerns, that need to be addressed:
- In Sec. 2.2 the thermo-mechanical model of the buried pipelines is employed for the evaluation of the vertical displacement due to the differential frost heave of soils. The thermal boundary conditions at the sides are assumed as insulation conditions, whereas thermal boundary conditions at bottom and upper sides are with fixed temperature. Under a such condition the temperature should be almost uniform in the direction of the pipe Moreover, these boundary conditions are not consistent with the Fig.1 where the frost condition is not uniform. This point should be clarified.
- The phrase “First, verify the temperature field of the thermal coupling model, which lays the foundation for the study of the stress field in the pipe-soil thermal-mechanical coupling model” at line number 181-183 is not clear.
- Figure numbers 6 and 7 should be 5 and 6.
- What do you mean with “encrypted” at line n. 159?
- Section 3 is: “Mechanical analysis of Inner Wall of buried non-destructive pipeline in seasonal frozen soil region.” The term “non-destructive” is usually used for test. Is it appropriate for such a case?
- In Sec. 2, at line number 94-95, is the term “calculated” right? Modelled could be better?

Author Response
Dear reviewer:
Thank you for your comments concerning our manuscript entitled “Analysis and evaluation on residual strength of pipelines with internal corrosion defects in seasonal frozen soil region” (applsci-1490962).
Dear reviewer:
Thank you very much for your comments and suggestions on the manuscript, which will help to improve the integrity and rationality of this article. Below we have pasted your comments in blue, our point-by-point response is given in black. Here, we will make corresponding explanations for the following questions.
1、In Sec. 2.2 the thermo-mechanical model of the buried pipelines is employed for the evaluation of the vertical displacement due to the differential frost heave of soils. The thermal boundary conditions at the sides are assumed as insulation conditions, whereas thermal boundary conditions at bottom and upper sides are with fixed temperature. Under a such condition the temperature should be almost uniform in the direction of the pipe axis. Moreover, these boundary conditions are not consistent with the Fig.1 where the frost condition is not uniform. This point should be clarified.
- Thank you for your reminder.
Fig. 1 shows the two-dimensional plane of the pipeline-soil location relationship along the axial direction of the pipeline without frost heaving. This may lead to misunderstanding of the initial boundary conditions. Therefore, we modify Fig. 1 to a two-dimensional plane as shown in applsci-1490962-revision. The thermal boundary condition of the side of the model was set to insulation condition, while the bottom was constant temperature and the upper thermal boundary condition was cyclic large temperature. In this way, with the change of surface temperature, differential frost heaving occurs in the soil along the axial direction of the pipeline. The specific boundary conditions and initial conditions are shown in Section 2.2 of the revised manuscript.
2、The phrase “First, verify the temperature field of the thermal coupling model, which lays the foundation for the study of the stress field in the pipe-soil thermal-mechanical coupling model” at line number 181-183 is not clear.
- Thank you for your reminder.
- Since the physical and mechanical properties of frozen soil are greatly affected by the temperature field, this model adopts the sequential coupling method. Firstly, the temperature field and actual temperature monitoring data of buried pipelines in frozen soil area under the influence of surface circulation atmospheric temperature are compared and verified to ensure that the model can accurately restore the real thermal state of soil and provide guarantee for the simulation of stress field.
- We rewrite this sentence in the manuscript as follows:
In order to ensure the accuracy of the stress field in the pipeline-soil thermo-mechanical coupling model, we first verified the temperature field of the coupling model.
- Figure numbers 6 and 7 should be 5 and 6.
- Thank you for your reminder.
- We have revised it in the manuscript.
- What do you mean with “encrypted” at line n. 159?
- “Encrypted” ——In order to get a more accurate answer, we divide the key part of the model into smaller grids.
- We have decided to revise “encrypted” to “grid refined”'. As follow:
As shown in Figure 3(b), the elements of the soil within the range of 5 m near the junction of the soil transition section and the frost heaving section and the soil near the axial direction of the pipeline should be grid refined, with a total of 28431 elements.
- Section 3 is: “Mechanical analysis of Inner Wall of buried non-destructive pipeline in seasonal frozen soil region.” The term “non-destructive” is usually used for test. Is it appropriate for such a case?
- Thank you for your advice. My word here is really not accurate enough. It should be changed to "non-corroded", which means the pipeline before it has not been corroded.
- In Sec. 2, at line number 94-95, is the term “calculated” right? Modelled could be better?
- Thank you for your valuable advice.
-The word we use here is really not accurate enough, “calculated” should be changed to “Modelled”. We have revised it in Section 2 of the manuscript.
7、The English language and style have been changed.
Please download the following file. Thank you!

Reviewer 3 Report
The authors evaluated the effects of internal corrosion defects on residual strength of buried pipeline made of X65 steel in consideration of seasonal temperature influences. Semi-empirical equations that predict residual strength of the pipeline was derived using the SPSS program and verified its effectiveness with international design code and burst data presented in a previous literature.
Reviewer considers that this manuscript has contributed to enhancement of pipeline integrity assessment, thus would like to suggest Editor to accept this paper for the publication, provided that some improvements are to be made by authors based on following comments .
- In Section 2.1.2 authors need to describe why temperature dependent material properties are not considered for pipeline and insulation material, although considered for soil.
- Line 136, element type used for heat transfer should not be C3D8. Besides it would be good to describe number of element and nodes created for the FE model.
- Add a brief introduction to Mohr-Coulomb elastoplastic constitutive model in somewhere in Section 2.
- Details of the contact condition need to be mentioned and additional explanation should be provided for the friction coefficient of 0.3 used for tangential behaviour.
- Define location of the two enlarged figure shown in Figure 3 (b).
- Line 188 and 190, figure numbers seem to be cited wrongly.
- It would be good to add an additional figure that shows temperature gradients adjacent to the pipeline.
- In Section 3 more descriptions are needed for boundary condition and loading condition applied to the FE model.
- It is hard to read lines and symbols in Figure 9.
- Give a brief introduction to Orthogonal Design Method.
- No need to start a section number with 4.2.1 unless 4.2.2 followed.
- Use unit of MPa rather than Mpa consistently in the manuscript.
Author Response

(The authors gave the same response as above.)
